# A New Cell Line Derived from the Caudal Fin of the Dwarf Gourami (*Trichogaster lalius*) and Its Susceptibility to Fish Viruses

**DOI:** 10.3390/biology12060829

**Published:** 2023-06-07

**Authors:** Ye-Jin Jeong, Kwang-Il Kim

**Affiliations:** Department of Aquatic Life Medicine, Pukyong National University, Busan 48513, Republic of Korea; 201513346@pukyong.ac.kr

**Keywords:** dwarf gourami, primary cell, infectious spleen and kidney necrosis virus, red sea bream iridovirus

## Abstract

**Simple Summary:**

Fish-derived primary cells are highly valuable for investigating cell–virus interactions, virus isolation, diagnosis, and vaccine development because of their rapid growth rate and high susceptibility to viruses. We established a new cell line, the dwarf gourami fin (DGF), from the caudal fin of the dwarf gourami (*Trichogaster lalius*). We analyzed the optimal growth conditions, including media, temperature, and fetal bovine serum concentration, modal chromosome number, and transfection efficiency. The susceptibility of DGF cells to different viruses, including red sea bream iridovirus, infectious spleen and kidney necrosis virus, viral hemorrhagic septicemia virus, hirame rhabdovirus, and spring viraemia of carp virus was evaluated.

**Abstract:**

The detection of megalocytiviruses, especially the infectious spleen and kidney necrosis virus (ISKNV), in ornamental fish has increased with the rapid growth of the ornamental fish industry. In this study, dwarf gourami fin (DGF) cells derived from the caudal fin of the dwarf gourami (*Trichogaster lalius*), which is highly susceptible to red sea bream iridovirus (RSIV) and ISKNV, were established and characterized. The DGF cells were grown at temperatures ranging from 25 °C to 30 °C in Leibovitz’s L-15 medium supplemented with 15% fetal bovine serum and were subcultured for more than 100 passages, predominantly with epithelial-like cells. DGF cells had a diploid chromosome number of 2*n* = 44. Although the initial purpose of this study was to establish a cell line for the causative agents of red sea bream iridoviral disease (RSIV and ISKNV), DGF cells were also susceptible to rhabdoviruses (viral hemorrhagic septicemia virus, hirame rhabdovirus, and spring viraemia of carp virus), exhibiting a significant cytopathic effect characterized by cell rounding and lysis. Additionally, viral replication and virion morphology were confirmed using virus-specific conventional polymerase chain reaction and transmission electron microscopy. Furthermore, both RSIV and ISKNV were replicated at high concentrations in DGF cells compared to other cell lines. Notably, the DGF cells maintained a monolayer during ISKNV infection, indicating the possibility of persistent infection. Thus, DGF can be used for viral diagnosis and may play a critical role in advancing our understanding of ISKNV pathogenesis.

## 1. Introduction

Ornamental fish have become a growing sector in the international fish trade, primarily because of the increasing popularity of household and public aquaria worldwide [1]. The ornamental fish trade is a multibillion-dollar industry estimated to involve more than 125 countries, and more than two billion live ornamental fish are traded annually [2]. Various pathogens have been detected during live ornamental fish trades, especially infectious spleen and kidney necrosis virus (ISKNV), which has been detected in various ornamental fish, including peal gourami (*Trichopodus leerii*), moonlight gourami (*Trichopodus microlepis*), dwarf gourami (*Trichogaster lalius*), southern platyfish (*Xiphophorus maculatus*), green swordtail (*Xiphophorus helleri*), molly (*Poecilia sphenops*), guppy (*Poecilia reticulatus*), oscar (*Astronotus ocellatus*), neon tetra (*Hyphessobrycon innesi*), and angel fish (*Pterophyllum eimekei*) [3].

Members of the genus *Megalocytivirus* (family Iridoviridae) comprise double-stranded DNA enclosed within an icosahedral capsid surrounded by an envelope, and are classified into four viruses: infectious spleen and kidney necrosis virus (ISKNV), red sea bream iridovirus (RSIV), turbot reddish body iridovirus (TRBIV), and scale drop disease virus (SDDV) [4]. Among these, RSIV and ISKNV are causative agents of red sea bream iridoviral disease (RSIVD), which has a broad host range and infects over 40 species of freshwater and marine fish, resulting in significant global economic losses [5]. ISKNV-infected ornamental fish have been reported in imported ornamentals in retail shops [6,7,8], and during international trade around the world [8,9]. Despite ongoing detection of ISKNV in ornamental fish and the growth of the industry, there is a lack of research on the control and treatment of this virus in ornamental fish. Dwarf gourami, a small ornamental fish popular for its vibrant colors and peaceful temperament, accounts for a high proportion of ISKNV-like megalocytivirus detections in Australia’s quarantine process [8,9]. Therefore, we aimed to develop a new cell line derived from dwarf gourami, an ornamental fish with high susceptibility to RSIV and ISKNV.

Cells are highly valuable for in vitro investigations of cell–virus interactions, virus isolation, diagnosis, and vaccine development because of their fast growth rate and high susceptibility to viruses. More than 783 fish-derived cells have been developed for research since in vitro culture of fish cells started [10]. In addition, cell lines that are highly permissive to viruses play an important role in virus research, such as vaccine development and virus infection mechanisms. Although several cell lines have been developed for RSIV and ISKNV research in recent years [11,12,13,14,15,16,17,18], the lack of cell lines derived from ornamental fish has limited our understanding of the mechanisms of ISKNV infection in ornamental fish despite the increasing importance of this industry. Therefore, the development of ornamental-fish-derived cell lines is important to understand and fill this knowledge gap.

This study aimed to establish and optimize the culture conditions of a new cell line derived from the caudal fin of dwarf gourami and to evaluate its susceptibility to fish pathogenic viruses. Specifically, we investigated the replication ability of two megalocytiviruses, RSIV and ISKNV, in a new cell line and compared it to that in other fish cell lines.

## 2. Materials and Methods

### 2.1. Primary Culture and Subculture of Dwarf Gourami Fin Cells

Healthy dwarf gourami (body length: 2.5 cm, body weight: 5.0 g) were obtained from an ornamental fish shop in Busan. Prior to experimentation, dwarf gourami was screened for fish viruses using PCR analysis. The primers used for the PCR are listed in Table 1. The fish was anesthetized with ice and wiped with 70% alcohol. The caudal fin was collected and washed once with phosphate-buffered saline (PBS; Gibco, Grand Island, NY, USA) containing a 3 × antibiotic-antimycotic solution (AA Gibco, Grand Island, NY, USA). The dissected tissues were cut into small pieces (approximately 1 mm^2^). These tissue fragments were then washed three times with PBS containing 3 × AA. The cells were subsequently dispersed by stirring with a 0.25% trypsin-EDTA solution (Gibco, Grand Island, NY, USA) supplemented with 3 × AA for 45 min. After the digestion, the cells were suspended in 10 mL of complete cell culture medium, Leibovitz’s L-15 medium (L-15 medium; Gibco, Grand Island, NY, USA) supplemented with 20% fetal bovine serum (FBS; Gibco, Grand Island, NY, USA) and 2 × AA. The cell suspension was then filtered through a cell strainer (pore size; 100 μm; Falcon, Midland, MI, USA). The filtered cells were centrifuged at 500× *g* for 10 min at 4 °C, and the supernatant was discarded. This process was repeated three times. Primary cultured cells were transferred into a 25 cm^2^ culture flask and maintained at 25 °C. The culture medium was replaced daily, and the morphology of the attached cells was observed under a microscope. The primary cultured cells were designated as dwarf gourami fin (DGF) cells.

After 7 days of primary culture, DGF cells attained over 90% confluence in a 25 cm^2^ cell culture flask, and were digested with 0.25% trypsin-EDTA, subcultured at a ratio of 1:3, and maintained in a complete cell culture medium. The subcultured cells were passaged every five days. Cultured cells were stored every 5–10 passages at −80 °C or −176 °C (liquid nitrogen) using cell banker 1 (Zenoaq, Koriyama, Fukushima, Japan).

### 2.2. Cell Growth Optimization

To determine the optimal conditions for DGF cell culture, DGF cells (5.0 × 10^4^ cells/well) were seeded in 12-well culture plates. The effects of the cell culture medium, temperature, and FBS concentration on cell growth were investigated. The cells were trypsinized, collected for counting at days 1, 3, 5, 7, and 9, and counted using a C-Chip disposable hemocytometer (INCYTO, Cheonan, Chungcheongnam-do, Republic of Korea). For the evaluation of culture medium, cells were cultured with L-15, Dulbecco’s modified Eagles Media with high glucose (DMEM-H; Welgene, Gyeongsan, Gyeongsanbuk-do, Republic of Korea), and low glucose (DMEM-L; Welgene, Gyeongsan, Gyeongsanbuk-do, Republic of Korea) containing 15% FBS, and were incubated at 25 °C. To find the optimal temperature, DGF cells were cultured with L-15 medium including 15% FBS and incubated at 20, 25, 28, and 30 °C. Lastly, cell growth evaluation was also determined in L-15 medium supplemented with different concentrations of FBS (2%, 5%, 10%, 15%) and incubated at 28 °C. All experiments were performed in triplicate, and statistical analysis was conducted using two-way analysis of variance (ANOVA) using GraphPad Prism (version 9.5.1). Statistical significance was set at *p* < 0.05.

### 2.3. Chromosome Analysis

Two days prior to the experiment, DGF cells were subcultured in a 25 cm^2^ cell culture flask. The cells were washed once with PBS containing 1 × AA, and colchicine solution (Merck, Darmstadt, Germany) was added at a final concentration of 0.5 μg/mL and incubated for 5 h at 25 °C. The cells were then collected by cell scraper and harvested by centrifugation at 500× *g* for 10 min at 4 °C. The supernatant was discarded and 0.07 M KCl was added and incubated at 25 °C for 25 min. Cells were pre-fixed by Carnoy’s solution (methanol–acetic acid 3:1) for 2 min and centrifuged at 500× *g* for 10 min at 4 °C. The supernatant was discarded and the cells were re-fixed with chilled Carnoy’s solution at room temperature for 25 min. This process was repeated twice. Then, the cells were resuspended with Carnoy’s solution and stored at −20 °C overnight. The cells were resuspended and dropped onto glass slides, which were precleaned with Carnoy’s solution and chilled. After drying, the cells were stained with 5% Giemsa stain (pH 6.8) for 25 min at room temperature. The staining solution was washed off, and the glass slide was dried. Metaphase spreads of chromosomes were observed under a light microscope (1000× magnification), and 100 metaphase spreads were counted.

### 2.4. Species Confirmation of DGF Cells

The origin of the DGF cell line was determined through the analysis of mitochondrial cytochrome c oxidase subunit 1 (COI) gene sequence. Genomic DNA was extracted from DGF cells using the Patho Gene-Spin^TM^ DNA/RNA Extraction Kit (iNtRON Biotechnology, Seongnam, Kyonggi-do, Republic of Korea) according to the manufacturer’s protocol. The COI gene was amplified using the following primers: DwGCOI-F:5′-GTC ACA GCA CAC GCT TTT GT-3′ and DwGCOI-R:5′-TGC TGG CTA GTG GAG GGT AT-3′. The PCR conditions were as follows: initial denaturation at 95 °C for 5 min; 30 cycles at 95 °C for 1 min, 60 °C for 1 min, and 72 °C for 1 min; and a final extension at 72 °C for 5 min. The PCR products were sequenced by Bionics using an ABI 3730XL DNA Analyzer (Applied Biosystems, Foster, CA, USA), and the nucleotide homology was confirmed using the National Center for Biotechnology Information (NCBI) website and the Basic Local Alignment Search Tool (BLAST) program (http://www.ncbi.nlm.nih.gov/BLAST, accessed on 1 June 2022). A phylogenetic tree was constructed using the maximum likelihood method with 1000 bootstrap values using MEGA software (ver.11.0.10).

### 2.5. Transfection of GFP Vector

DGF cells were passaged into a confocal dish at a density of 3.0 × 10^5^ cells/mL. After being cultured for 20 h at 28 °C, DGF cells were transfected with pAcGFP1 vector (Takara, Kusatsu-Shi, Chuo-Ku, Japan) using Lipofectamine 3000 reagent and pAcGFP1 vector. A total of 5 µg of pAcGFP1 vector was diluted by 250 µL of Opti-MEM medium (Gibco, Grand Island, NY, USA). The diluted pAcGFP1 vector was mixed with Lipofectamine 3000 and incubated at RT for 15 min. This mixture was then added to a confocal dish. After 48 h, green fluorescence signals were detected using a Ts2 FL (Nikon, Minato-Ku, Kadoma-Shi, Japan).

### 2.6. Propagation of Viruses

#### 2.6.1. Virus Susceptibility and Viral Multiplication

Various fish viruses were inoculated into DGF cells to evaluate their susceptibility to viruses. The viruses used in this experiment are as follows: red sea bream iridovirus (RSIV, 17RbGs strain; isolated from rock bream; [25]), infectious spleen and kidney necrosis virus (ISKNV, PGIV strain; isolated from pearl gourami; [7]), viral hemorrhagic septicemia virus (VHSV; isolated from olive flounder; [26]), hirame rhabdovirus (HIRRV; ATCC, VR-1391), and spring viraemia of carp virus (SVCV, SVCV-K1 strain; isolated from common crap; [27]); nervous necrosis virus (NNV, RGNNV-type; isolated from sea bass; [26]). The infectivity titer of the virus was measured by TCID_50_ assay using DGF, epithelioma papulosum cyprini (EPC), and E11 cell lines for megalocytiviruses (RSIV and ISKNV), rhabdoviruses (VHSV, HIRRV, and SVCV), and NNV, respectively.

A viral inoculum with a concentration of 10^5.8^ TCID_50_/mL was adsorbed onto a monolayer of DGF cells in a 25 cm^2^ cell culture flask, and incubated for 2 h. The cells were then washed twice with PBS and cultured in 5% heat-inactivated FBS (at 55 °C for 30 min) supplemented L-15 medium (L-15_5_ medium). Incubation temperatures for RSIV-, ISKNV-, and NNV-inoculated DGF cells were 28 °C, while those for other viruses (VHSV, HIRRV, and SVCV) were 20 °C. The cytopathic effect (CPE) was observed daily under a microscope. The supernatants were harvested either upon the appearance of CEP or at 14 days post-inoculation (dpi), followed by three freeze–thaw cycles, and subsequently centrifuged at 500× *g* for 10 min at 4 °C. Total DNA or RNA was extracted using the Patho Gene-spin^TM^ DNA/RNA Extraction Kit, according to the manufacturer’s instructions. To detect RNA viruses (rhabdovirus and NNV), cDNA was synthesized using the PrimeScript^TM^ RT Reagent Kit (Takara, Shiga, Japan) following the manufacturer’s instructions. Cultured viruses were identified by PCR. The primers and PCR conditions used in this study are listed in Table 1. After confirming virus susceptibility, DGF-formed monolayers in 24-well culture plates were infected with RSIV, ISKNV, and rhabdoviruses using the same method as mentioned above. RSIV- and ISKNV-inoculated DGF cells were incubated at 28 °C, and rhabdovirus-inoculated DGF cells were incubated at 20 °C for 7 days. At 7 dpi, the supernatants were collected, and each supernatant was quantified using the TCID_50_ assay: RSIV and ISKNV with DGF cells, and VHSV, HIRRV, and SVCV with EPC cells throughout this study.

#### 2.6.2. Transmission Electron Microscopy

DGF cells infected with megalocytiviruses (RSIV and ISKNV) and rhabdoviruses (VHSV, HIRRV, and SVCV) were collected and pre-fixed in 2.5% glutaraldehyde buffer at 4 °C overnight. The cells were subsequently immobilized in 1% osmium tetroxide, dehydrated using graded ethyl alcohol, and embedded in epoxy resin for sectioning. The sections were double-stained with uranyl acetate and lead citrate and examined under a transmission electron microscope (JEM-1200EXII, JEOL).

### 2.7. Characteristics of RSIV and ISKNV Replication in DGF Cells

#### 2.7.1. Extracellular Virus Propagated in DGF Cells

DGF cells that had formed a monolayer in a 24-well cell culture plate were infected with RSIV and ISKNV-type viruses using the same method mentioned in Section 2.6.1 and incubated at 28 °C for 11 days. The supernatant of the culture media was collected from each well at 1, 3, 5, 7, 9, and 10 dpi to measure the infectivity titers and genome copies of propagated RSIV and ISKNV in the DGF cells. An infectivity test was performed using the TCID_50_ assay in DGF cells. The genome copies of RSIV and ISKNV were determined via real-time PCR using the method described by Kim et al., (2021) [20]. Briefly, real-time PCR was performed using a 20 μL (total volume) mixture containing 1 μL of DNA extracted from each virus, 200 nM of each primer (RSIV 1094F/1221R) and probe (RSIV 1177 probe) (Table 1), 10 μL of 2× HS Prime qPCR Premix (Genet Bio, Yuseong-gu, Daejeon, Republic of Korea), 0.4 μL of 50× ROX dye, and 7.1 μL of DEPC-treated water. Amplification was performed using a StepOne Real-time PCR System (Applied Biosystems, Waltham, MA, USA) with cycling conditions of 95 °C for 10 min, followed by 40 cycles of 94 °C for 10 s and 60 °C for 35 s.

#### 2.7.2. Comparison of DGF and Other Cell Lines for Megalocytivirus Propagation

DGF, grunt fin (GF), *Pagrus major* fin (PMF [18]), and rock bream fin (RBF [28]) cells were used. The cells were grown to form a monolayer in a 24-well cell culture plate and were infected using the method mentioned in Section 2.6.1. and incubated at 28 °C for 7 days. The supernatants were collected at 7 dpi and the infectivity titers and genome copies were measured as described above. The infectious viral titer was analyzed by two-way analysis of variance (ANOVA) using GraphPad Prism (version 9.5.1). Statistical significance was set at *p* < 0.05.

#### 2.7.3. Viral Gene Expression

To determine the difference in propagation between RSIV and ISKNV, the megalocytivirus viral gene was analyzed using four viral genes: major capsid protein (MCP), adenosine triphosphatase (ATPase), polymerase, and myristoylated membrane protein (MMP) genes. The viral inoculation was performed as described in Section 2.6.1. Total RNA was extracted from virus-inoculated cells at 6, 24, 48, and 72 h using the yesR^TM^ Total RNA Extraction Kit (GenesGen, Busan, Republic of Korea). cDNA synthesis was performed using the PrimeScript^TM^ RT Reagent Kit, following the manufacturer’s protocol. Real-time PCR reactions were conducted in a total volume of 20 µL, comprising 10 µL of Prime Q-master mix (SYBR GREEN1; Genet Bio), 1 µL of cDNA, 0.5 µL of each primer set, and 8 µL of DEPC-treated water. The primer sets and amplification protocols used in this study are listed in Table 2. Each gene expression analysis was performed in triplicate. The β-actin gene was chosen as the housekeeping gene. Relative mRNA expression was determined using the 2^−∆∆CT^ method.

## 3. Results

### 3.1. Development of DGF Cells

The division of primary cells was observed in close proximity to fragments from the dwarf gourami caudal fin on the second day, as shown in Figure 1A. By the seventh day at 25 °C, a monolayer consisting of epithelial-like and fibroblast-like cells had formed (Figure 1B). In the early passages (up to 10 passages), the cells were subcultured at a ratio of 1:3 every 7–9 days and cultured in L-15 medium supplemented with 20% FBS and 2 × AA. After subculturing for more than 10 passages, the concentrations of FBS and AA were reduced to 15% and 1 ×, respectively. As passage time increased (over 40 passages), it was observed that epithelial-like cells were predominant (Figure 1C); the FBS concentration was further reduced to 10%, and 5 mL of MEM non-essential amino acid solution (NEAA; Gibco, Grand Island, NY, USA) and 10 mL of HEPES (Gibco, Grand Island, NY, USA) were added to the cell culture medium. Currently, the cell line has been subcultured for over 100 passages and is designated as the DGF cell line (Figure 1D). Cryopreserved cells were successfully recovered and formed a monolayer in a 25 cm^2^ cell culture flask within five days. DGF cells at passage 96 were deposited in the Korean Cell Line Research Foundation (KCLRF) under the accession number KCLRF-00522.

### 3.2. Species Confirmation of DGF Cells

The origin species of the DGF cell line was verified by analyzing its COI gene sequence. The COI gene was amplified from the extracted cellular DNA, and the expected PCR product of 243 bp was obtained. The amplified sequence exhibited 100% identity with the dwarf gourami gene (GenBank accession No. KU569055) and belonged to the same branch (Figure 2), confirming that the DGF cells originated from dwarf gourami.

### 3.3. Chromosome Number

Chromosomal analysis was performed to investigate the chromosomal characteristics of the DGF cell line. The number of chromosomes in the DGF cells varied from 35 to 49, with a modal number of 2*n* = 44 (Figure 3).

### 3.4. Growth Characteristics of DGF Cells

To determine the optimal growth conditions for the newly established DGF cells in vitro, we investigated the culture medium, incubation temperature, and FBS concentration. Successful cell growth was only observed when using L-15 medium containing 15% FBS, whereas DMEM showed no ability to grow DGF cells, causing them to detach from the well plates regardless of glucose concentration (Figure 4A). Therefore, the DGF cells were maintained in L-15 medium containing 15% FBS. The results of the temperature-dependent growth analysis suggest that cells could be maintained at 20 °C without growing, but they could grow at a temperature ranging from 25 °C to 30 °C, with the most efficient growth observed at 30 °C (Figure 4B). In addition, at 28 °C, explosive growth occurred between 5 d and 7 d, and by day 9, there was no significant difference in rate compared to growth at 30 °C. These results suggest that the optimal growth temperature for DGF cells is 28 °C to 30 °C. When cells were maintained and grown in L-15 medium supplemented with 5%, 10%, and 15% FBS (Figure 4C), they showed explosive growth in 15% FBS-containing L-15 medium at 9 d. During the initial 1 to 5 d of culture, there was no significant difference in growth between cells cultured in 10% and 15% FBS-containing L-15 medium.

### 3.5. DGF Cell Transfection

The pAcGFP1 plasmid was successfully transfected into DGF cells using Lipofectamine 3000 and green fluorescence was observed at 24 h post-transfection. More than 30% of the cells expressed the fluorescence signals, as shown in Figure 5.

### 3.6. Virus Susceptibility

DGF cells inoculated with megalocytiviruses (RSIV and ISKNV), rhabdoviruses (VHSV, HIRRV, and SVCV), and NNV to test their virus susceptibility exhibited CPE (Figure 6) and specific PCR amplicons were generated. CPE, characterized by cell rounding and enlargement, was observed within 2–3 days of inoculation with RSIV and ISKNV (Figure 6A,B). Although all RSIV-inoculated cells shrunk and detached from the plate within 8–10 dpi, interestingly, no cell shrinkage or detachment was observed in ISKNV-inoculated cells. The effects of rhabdovirus infection (VHSV, HIRRV, and SVCV) on DGF cells were not clearly observed initially. However, after 5–7 dpi, the characteristic CPE of cell lysis was observed in rhabdovirus-inoculated DGF cells (Figure 6C–E). Although CPE appeared to slow down by day 10, no cells adhered to the cell culture plate. In contrast, no cytopathic effect was observed in NNV-inoculated DGF cells, and no specific PCR amplicons were generated from the supernatant. The replication of megalocytiviruses (RSIV and ISKNV) and rhabdoviruses (VHSV, HIRRV, and SVCV) in DGF cells was also confirmed by electron microscopy. Infected DGF cells showed a large number of viral particles with icosahedral nucleocapsids, a characteristic morphology of megalocytiviruses (Figure 7A,B). Additionally, only a small number of bullet-shaped viral particles were observed in the cytoplasm of rhabdovirus-infected DGF cells (Figure 7C–E).

### 3.7. Viral Multiplication

The amounts of megalocytiviruses (RSIV and ISKNV) and rhabdoviruses (VHSV, HIRRV, and SVCV) released from the infected DGF cells were determined using the TCID_50_ assay after 7 dpi. As shown in Figure 8, RSIV exhibited the highest multiplication, reaching 10^9.29^TCID_50_/mL. A similar infectivity titer was observed for ISKNV and HIRRV as 10^8.48^TCID_50_/mL and 10^8.44^TCID_50_/mL, respectively. Among the five viruses tested, VHSV and SVCV showed the lowest replication titers of 10^7.74^TCID_50_/mL and 10^7.72^TCID_50_/mL, respectively, despite the high propagation concentration. These results suggest that all five viruses can be replicated at high concentrations by DGF cells.

### 3.8. Propagation of RSIV and ISKNV

The genome copy and infectivity titer of propagated viruses was titered by real-time PCR and TCID_50_ assay, respectively. As shown in Figure 9, in RSIV-inoculated DGF cells, the genome copy and infectious viral titers in the supernatant were markedly increased between 3 dpi (8.54 × 10^6^ genome copies/mL, 10^4.39^ TCID_50_/mL) and 5 dpi (5.15 × 10^8^ genome copies/mL, 10^6.72^ TCID_50_/mL), and steadily increased up to 11 dpi when the RSIV-type release reached a plateau (2.50 × 10^9^ genome copies/mL, 10^9.08^ TCID_50_/mL). In DGF cells inoculated with ISKNV, the genome copy number was higher than that in RSIV-inoculated cells at 1 dpi, but the infectivity titer was lower. The genome and infectivity titers of RSIV remained higher than those of ISKNV throughout the experiments. The genome copies and infectivity titers in the supernatant of ISKNV-inoculated DGF cells markedly increased between 3 dpi (1.36 × 10^6^ genome copies/mL, 10^5.5^ TCID_50_/mL) and 5 dpi (3.24 × 10^8^ genome copies/mL, 10^7.17^ TCID_50_/mL). The genomic copy number was highest at 11 dpi (7.36 × 10^8^ genome copies/mL, 10^7.83^ TCID_50_/mL). However, the highest infectivity titer of ISKNV was observed at 7 dpi (2.22 × 10^8^ genome copies/mL, 10^8.06^ TCID_50_/mL).

### 3.9. Comparison of DGF and Other Cell Lines for RSIV and ISKNV Propagation

Within 7 days of RSIV inoculation, the typical CPE of megalocytivirus, such as cell rounding, shrinking, and detachment, was observed in DGF, RBF, and GF cells, whereas PMF cells showed only cell rounding (Figure 10A). In contrast, weak CPE in the form of rounded cells was observed in ISKNV-inoculated GF cells within 7 days, but no CPE was observed in ISKNV-inoculated PMF cells (Figure 10B). Notably, numerous enlarged rounding-up cells were observed in the RBF and DGF cells after 4 dpi (Figure 10B). Specifically, ISKNV caused cell dropout and monolayer disruption in RBF cells; however, monolayer disruption was not observed within 7 days in ISKNV-inoculated DGF cells. The supernatant of the infected cells was collected at 7 dpi, and both the infectivity and genomic titers were quantified (Figure 10C). For RSIV, the highest titers were found in PMF cells (5.98 × 10^10^genome copies/mL, 10^9.44^TCID_50_/mL); these were approximately five times higher than that in DGF cells (1.54 × 10^10^genome copies/mL, 10^8.74^TCID_50_/mL), but there was no statistical difference in the infectious viral titer. The highest ISKNV titers were found in DGF cells (5.22 × 10^9^ genome copies/mL, 10^8.48^ TCID_50_/mL). The titer of ISKNV in RBF cells was approximately 100-fold lower (1.88 × 10^7^genome copies/mL, 10^6.2^ TCID_50_/mL) than that in DGF cells. GF cells showed the lowest titers for both RSIV (1.20 × 10^8^ genome copies/mL; 10^7.03^ TCID_50_/mL) and ISKNV (3.27 × 10^5^ genome copies/mL, 10^4.36^ TCID_50_/mL).

### 3.10. Viral Gene Expression

The expression of four viral genes, MCP, ATPase, polymerase, and MMP, was analyzed in DGF cells (Figure 11). The MCP gene was expressed later but showed the highest expression at 72 h compared to the other genes, whereas ATPase was expressed rapidly. Other genes showed a steady increase with infection over time. RSIV had higher viral gene expression levels than ISKNV, which aligns with the observation of higher replication of RSIV than ISKNV.

## 4. Discussion

ISKNV, a species of megalocytivirus, has been consistently detected in various ornamental fish, both in retail shops [6,7,8] and during global trade [8,9], leading to significant economic losses. However, studies on the control and treatment of ISKNV infections in ornamental fish are lacking. Therefore, this study aimed to establish a cell line derived from the dwarf gourami, an ornamental fish species susceptible to RSIV and ISKNV, to facilitate further research on megalocytivirus infections and their control in ornamental fish.

In this study, a new cell line derived from the caudal fin of the dwarf gourami (*Trichogaster lalius*) was established and characterized for potential applications in fish virus research. DGF cells were subcultured more than 100 times, and sequence analysis of the COI gene confirmed that the DGF cells were derived from dwarf gourami. In addition, the results indicated that the DGF cell line could express exogenous genes via transfection and could be used for the functional verification of genes in vitro. The chromosome number is an important characteristic for the identification of certain species. However, fish cell lines may not always contain the same number of chromosomes as the intact host species. For DGF cells, the modal value of the chromosome number was 2*n* = 44 (Figure 3), whereas the diploid chromosome number of the intact species was 2*n* = 46 [30]. The cell lines derived from the Japanese flounder (*Paralichthys olivaceus*) developed in a previous study did not exhibit the same chromosome number as the complete host species. For example, the Japanese flounder spleen (JFSP) cell line has a chromosome number of 2*n* = 68 [31], and the flounder fin (FFN) and flounder spleen (FSP) cell lines have 64 and 62 chromosomes, respectively [32], while the Japanese flounder brain (JFB) cell line has 48, which is identical to the Japanese flounder (2*n* = 48) [33]. The different chromosome numbers between dwarf gourami and DGF cells could be attributed to chromosomal instability resulting from long-term cell culture and processes, such as doubling, cleavage, or regrouping [32].

To optimize the conditions for DGF cell incubation, three culture media were evaluated: DMEM-high glucose, DMEM-low glucose, and L-15. Our results showed that DGF cells were unable to grow in DMEM and even detached from the well plate, regardless of the glucose concentration (Figure 4A). Conversely, L-15 medium, which has been widely used for over 30 years, contains a rich supply of amino acids, including glycine, arginine, histidine, and threonine, and can support the growth of DGF cells as well as other teleost cell lines [10,34]. Moreover, we observed an increased rate of DGF cell growth without any change in cell morphology after passage 40 when NEAA was added to the culture medium and buffered with HEPES. These findings suggested that high amino acid nutrition is essential for DGF cell culture. Interestingly, DGF cells showed growth at temperatures ranging from 25 to 30 °C, and were also able to maintain at 20 °C (Figure 4B). This may be due to the ability of poikilotherms such as fish, reptiles, amphibians, and invertebrates to tolerate a wide range of temperatures [34]. Other studies on ornamental fish cell lines, such as the giant gourami (*Osphronemus goramy*) spleen (GP) cell line [17], parrot fish brain, spleen, and heart (PFB, PFS, PFH; [35]) cell line, and *Aequidens rivulatus* brain (ARB8; [16]), have also reported optimal growth at temperatures similar to those observed in our study.

Cell lines are considered ideal tools for studying cell–virus interactions, virus isolation, diagnosis, and vaccine development in vitro. Although the initial purpose of this study was to establish a cell line for the causative agents of RSIVD (RSIV and ISKNV), DGF cells were also susceptible to rhabdoviruses (VHSV, HIRRV, and SVCV). DGF cells inoculated with these viruses showed typical cytopathic effects (Figure 6C–E), and transmission electron microscopy confirmed the presence of intracellular virions (Figure 7C–E). However, it is interesting that DGF cells are susceptible to rhabdoviruses, while, currently, there is no information available on the susceptibility of dwarf gourami to rhabdoviruses. It is known that rhabdoviruses such as VHSV, HIRRV, and SVCV are typically pathogenic at low water temperatures ranging from approximately 3 to 17 °C [36]. Therefore, it is unlikely that infection studies have been conducted on dwarf gourami residing at higher temperatures exceeding 23 °C. Therefore, further assessment of the susceptibility of dwarf gourami to rhabdoviruses may be necessary.

Our findings confirmed that DGF cells are highly susceptible to both RSIV and ISKNV and show a higher replication capacity than other fish cell lines, such as GF, RBF [28], and PMF [18]. CPE was observed in cells that had been inoculated with RSIV or ISKNV (Figure 10A,B). At 7 dpi, cell rounding, enlargement, shrinkage, and detachment were observed in DGF, RBF, and GF cells infected with RSIV. In contrast, PMF cells exhibited strong cell rounding without detachment. Previous studies have shown that PMF cells have a characteristic advantage in persistently replicating at high concentrations of RSIV, without cell detachment [18]. Similarly, DGF did not show disruption of the monolayer, but strong cell rounding was evident after ISKNV infection with high propagation of ISKNV (5.22 × 10^9^ genome copies/mL corresponding to 10^8.48^ TCID_50_/mL). These results suggested that DGF cells have the potential to support persistent ISKNV infections. CPE is important for virus isolation and infectivity titration, such as TCID_50_ and plaque assays, because it is a visible manifestation of virus-induced damage to cells. However, GF cells, which are recommended by the World Organization for Animal Health for the isolation of RSIVD, do not form a complete CPE when inoculated with RSIV and require special techniques to maintain a steady viral titer [37]. For these reasons, several fish-derived cell lines were developed in recent decades to facilitate the study of megalocytiviruses. The mandarin fish fry (MFF-1; [11]) cell line has demonstrated efficient replication of ISKNV, reaching high concentrations of approximately 10^7.6–8.4^ TCID_50_/mL. Similarly, the spotted knifejaw (*Oplengathus punctatus*; SKF-9; [15]) cell line has been established, although it exhibits a lower propagation capability for ISKNV, with viral concentrations of approximately 10^6.5^ TCID_50_/mL achieved within 8 dpi, compared to approximately 10^5.1^ TCID_50_/mL for ISKNV. Compared to various previously developed cell lines, DGF cells propagated high concentrations for both RSIV and ISKNV. Therefore, DGF cells represent a valuable resource for the isolation and characterization of RSIV and ISKNV.

From the comparison of viral genome copies and infectivity titers in different susceptible cells, the highest copy numbers of RSIV and ISKNV were detected in PMF (5.98 × 10^10^ genome copies/mL, 10^9.44^ TCID_50_/mL) and DGF cells (5.22 × 10^9^ genome copies/mL, 10^8.48^ TCID_50_/mL), respectively (Figure 10C). There was no statistically significant difference in the infectivity titers between PMF and DGF cells (10^8.74^ TCID_50_/mL) during RSIV propagation. Additionally, the difference in infectivity titers between RSIV and ISKNV was insignificant (approximately 1.81 times) in DGF cells, whereas it was significant in RBF, PMF, and GF cells, at approximately 30.44 times, 113,588.02 times, and 469.61 times, respectively. This suggests that the cells exhibit different levels of susceptibility to RSIV and ISKNV. From the expression of viral genes including MCP, ATPase, polymerase, and MMP, the expression levels of all four genes were consistently higher in RSIV-infected cells than in ISKNV-infected cells (Figure 11). MCP and ATPase are considered conserved regions that have both been used for genotyping based on the phylogeny and molecular diagnosis of the megalocytivirus genus [38]. Polymerase also plays an important role in the replication of viral nucleic acids. Finally, MMP is a protein present in the virus envelope that plays a crucial role in the structure of the virus particle and the infection process [39]. Thus, the different expressions of major viral genes indicate differences in the replication and propagation of RSIV and ISKNV in DGF cells, which could be utilized as a useful tool for understanding RSIV and ISKNV infection mechanisms in future studies.

## 5. Conclusions

In summary, we successfully established a cell line called DGF, which was derived from the caudal fin of dwarf gourami. DGF cells were susceptible to megalocytiviruses (RSIV and ISKNV) and rhabdoviruses (VHSV, HIRRV, and SVCV). The broad susceptibility of DGF cells makes them excellent tools for studying various viruses, particularly those related to viral diseases in ornamental fish. These findings may lead to the development of effective strategies for the diagnosis, prevention, and control of viral diseases in ornamental fish.

## Figures and Tables

**Figure 1 biology-12-00829-f001:**
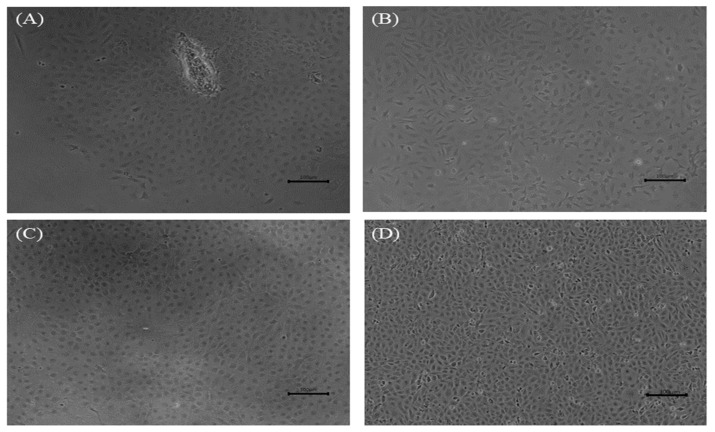
Morphology of the dwarf gourami fin (DGF) cells. (**A**–**D**) Morphology of DGF cells at primary passage on day 2 (**A**), and on day 7 (**B**), passage 63 (**C**), and passage 100 (**D**). Scale bar = 100 μm.

**Figure 2 biology-12-00829-f002:**
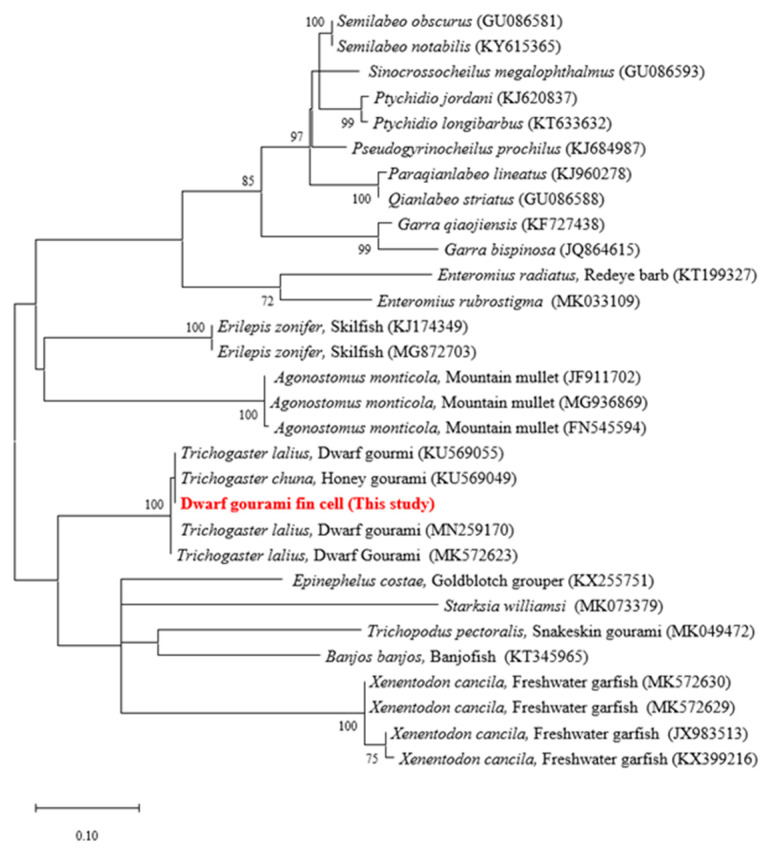
Phylogenic analysis to identify the mitochondrial cytochrome c oxidase subunit I (COI) gene of dwarf gourami fin (DGF) cells. The phylogenetic tree was constructed using the maximum likelihood algorithm with 1000 bootstrap replicates using MEGA software (ver.11.0.10). The COI gene from the DGF cell in this study is highlighted in bold and red.

**Figure 3 biology-12-00829-f003:**
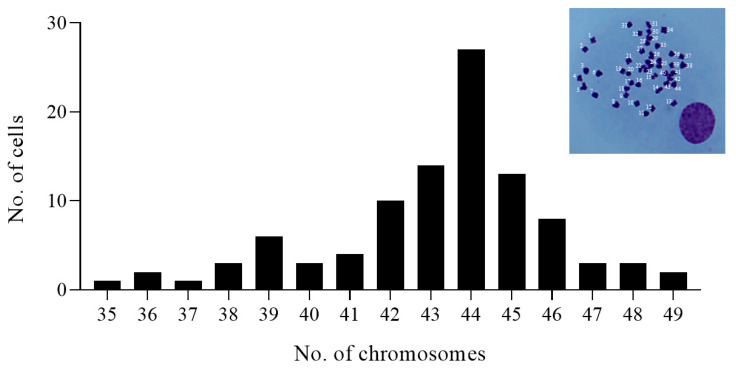
Chromosome analysis of the dwarf gourami fin (DGF) cells. Chromosome number and distribution of metaphase DGF cells, and representative picture for chromosomes of the DGF cell.

**Figure 4 biology-12-00829-f004:**
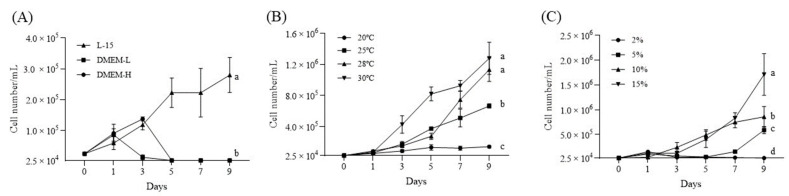
Growth curves of dwarf gourami fin (DGF) cells. (**A**) Number of DGF cells in various cell culture media (Leibovitz’s L-15 medium, L-15; Dulbecco’s modified Eagles Media with high glucose, DMEM-H; and Dulbecco’s modified Eagles Media with low glucose, DMEM-L) containing 15% fetal bovine serum (FBS) at 25 °C, (**B**) number of DGF cells in L-15 medium containing 15% FBS at different temperatures (20 °C, 25 °C, 28 °C, and 30 °C), and (**C**) number of DGF cells in L-15 medium containing different FBS concentrations (2%, 5%, 10%, and 15%) at 28 °C. The statistical analysis via two-way ANOVA was conducted using GraphPad Prism version 9.5.1. The cell numbers are expressed as the mean ± S.D. The significant differences (*p* < 0.05). The significant difference (*p* < 0.05) between groups are indicated by different lowercase letters (a, b, c, and d).

**Figure 5 biology-12-00829-f005:**
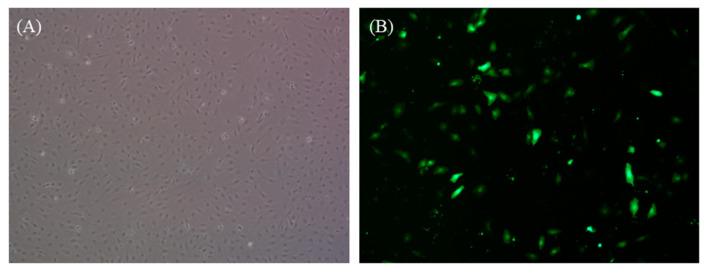
The expression of the green fluorescence protein gene in DGF cells 24 h after pAcGFP1 vector transfection in a confocal dish. (**A**) DGF cells under a bright field, (**B**) expression of GFP in DGF cells under a fluorescence microscope.

**Figure 6 biology-12-00829-f006:**
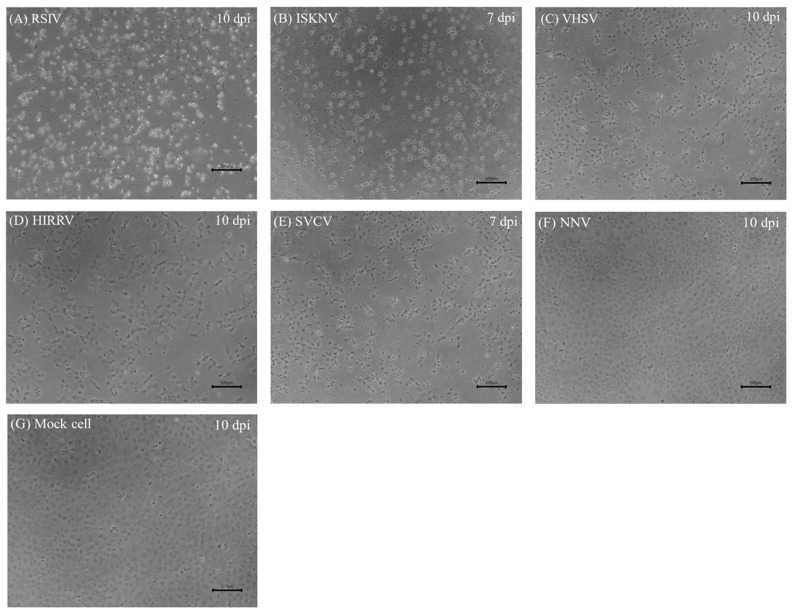
Infection of DGF cells with various fish viruses. Cytopathic effect (CPE) was observed under an inverted microscope. CPE of DGF cells infected with red seabream iridovirus (RSIV; (**A**)), infectious spleen and kidney necrosis virus (ISKNV; (**B**)), viral hemorrhagic septicemia virus (VHSV; (**C**)), hirame rhabdovirus (HIRRV; (**D**)), spring viraemia of carp virus (SVCV; (**E**)), nervous necrosis virus (NNV; (**F**)), and control (**G**). Scale bar = 100 μm.

**Figure 7 biology-12-00829-f007:**
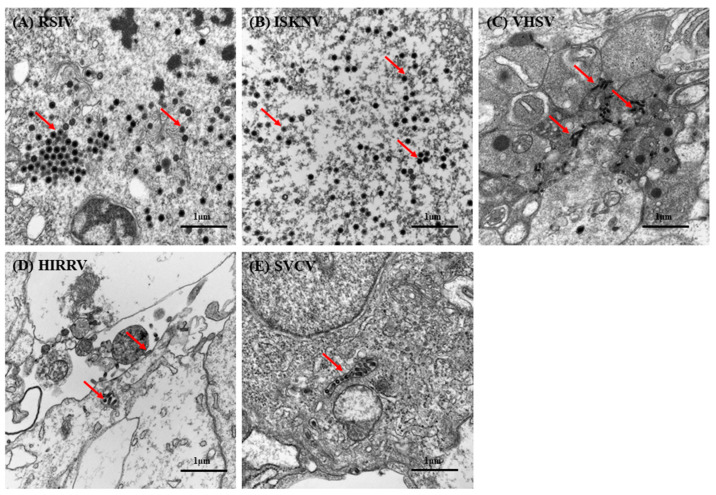
Transmission electron micrograph of virus-infected DGF cells under 30,000× magnification. (**A**) Red sea bream iridovirus (RSIV)-infected DGF cells, (**B**) infectious spleen and kidney necrosis virus (ISKNV)-infected DGF cells, (**C**) viral hemorrhagic septicemia virus (VHSV)-infected DGF cells, (**D**) hirame rhabdovirus (HIRRV)-infected DGF cells, (**E**) spring viraemia of carp virus (SVCV)-infected DGF cells. Arrows indicate virus particles. Scale bar = 1 μm.

**Figure 8 biology-12-00829-f008:**
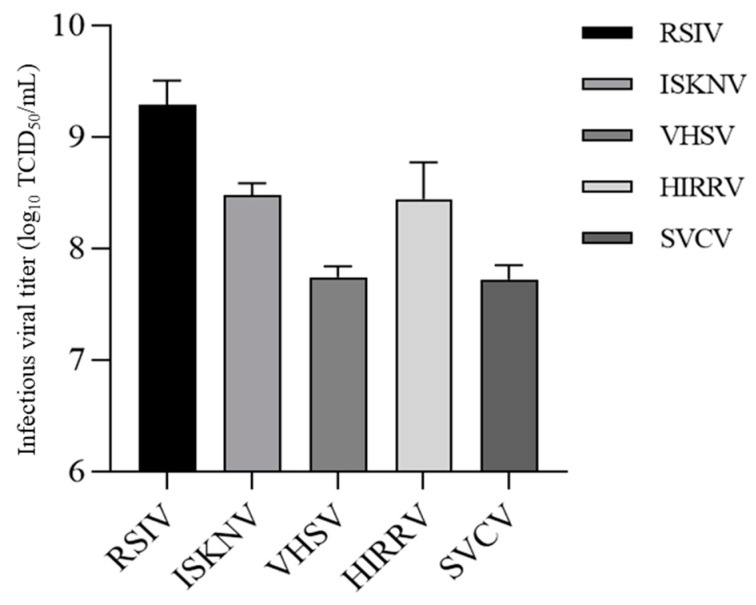
Propagation of megalocytiviruses (red sea bream iridovirus; RSIV and infectious spleen and kidney necrosis virus; ISKNV) and rhabdoviruses (viral hemorrhagic septicemia virus; VHSV, hirame rhabdovirus; HIRRV and spring viraemia of carp virus; SVCV) in DGF cells. Infectious viral titers from DGF cells inoculated with megalocytiviruses (RSIV and ISKNV) and rhabdoviruses (VHSV, HIRRV, and SVCV) at 7 days post-inoculation. The initial virus inoculum concentration was 10^5.8^ TCID_50_/mL. Megalocytivirus-inoculated DGF cells were incubated at 28 °C and rhabdovirus-inoculated DGF cells were incubated at 20 °C.

**Figure 9 biology-12-00829-f009:**
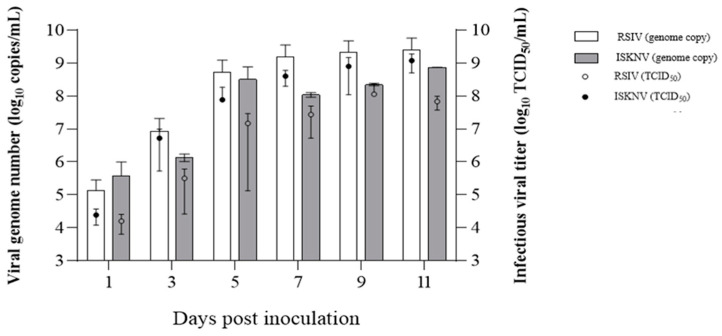
Infectivity titer and genome copy numbers of dwarf gourami fin (DGF) cells inoculated with red sea bream iridovirus (RSIV) and infectious spleen and kidney necrosis virus (ISKNV) at 1, 3, 5, 7, 9, and 11 days post-inoculation (dpi). The initial concentration of inoculum was 10^5.8^ TCID_50_/mL and virus-inoculated cells were incubated at 28 °C.

**Figure 10 biology-12-00829-f010:**
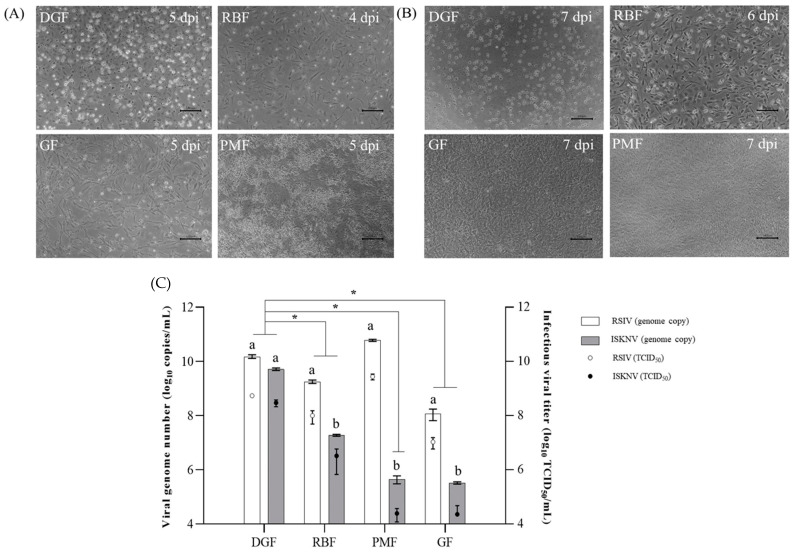
Comparison of cytopathic effect of red sea bream iridovirus- and infectious spleen and kidney necrosis virus-inoculated cells. (**A**) Cells infected with RSIV, and (**B**) cells infected with ISKNV. Scale bar = 100 μm. (**C**) Comparison of infectious viral titer and genome copy number of RSIV- and ISKNV-inoculated cells. The initial inoculum concentration was 10^5.8^ TCID_50_/mL. Virus-inoculated dwarf gourami fin (DGF) cells and rock bream fin (RBF) cells were incubated at 28 °C, while Pagrus major fin (PMF) cells and grunt fin (GF) cells were incubated at 25 °C for 7 days. The statistical analysis via two-way ANOVA was conducted using GraphPad Prism version 9.5.1. The infectious titer (TCID_50_ value) was expressed as the mean ± S.D. The significant differences (*p* < 0.05) among cell lines are indicated by different lowercase letters (a and b), and significant differences between cell lines are indicated by *, *p* < 0.05.

**Figure 11 biology-12-00829-f011:**
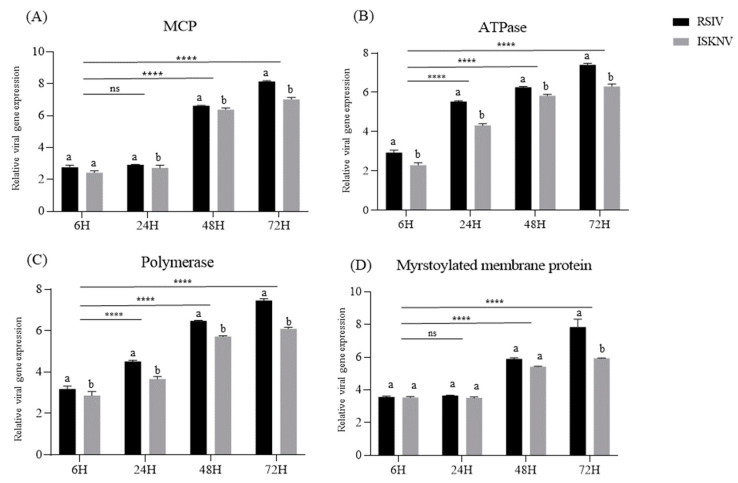
Comparison of viral gene expression in DGF cells inoculated with RSIV and ISKNV at a concentration of 10^5.8^TCID_50_/mL. The supernatant was sampled, respectively, at 6, 24, 48, and 72 h post-inoculation, and the viral genes were measured by SYBR green-based real-time PCR. (**A**) Major capsid protein (MCP), (**B**) adenosine triphosphatase (ATPase), (**C**) polymerase, and (**D**) myristoylated membrane protein genes. The statistical analysis via two-way ANOVA was conducted using GraphPad Prism version 9.5.1. The viral gene expression was expressed as the mean ± S.D. The significant differences (*p* < 0.05). The significant differences (*p* < 0.05) among groups are indicated by different lowercase letters (a and b), and significant differences between groups are indicated by ****, *p* < 0.05 and ns, not significant.

**Table 1 biology-12-00829-t001:** Primers used in this study.

Target Virus	Object	Primers	Sequence	Reference
Megalocytiviruses ^a^ (RSIV and ISKNV)	Detection	1-F	CTCAAACACTCTGGCTCATC	[19]
		1-R	GCACCAACACATCTCCTATC	
	Detection and quantification	RSIV 1094F	CCAGCATGCCTGAGATGGA	[20]
		RSIV 1221R	GTCCGACACCTTACATGACAGG	
		RSIV 1177 Probe	FAM-TACGGCCGCCTGTCCAACG-BHQ1	
NNV ^b^	Detection	VNN1	ACACTGGAGTTTGAAATTCA	[21]
		VNN2	GTCTTGTTGAAGTTGTCCCA	
VHSV ^c^	Detection	VHSV 3F	GGGACAGGAATGACCATGAT	[22]
		VHSV 2R	TCTGTCACCTTGATCCCCTCCAG	
HIRRV ^d^	Detection	HRV-F	ACCCTGGGATTCCTTGATTC	[23]
		HRV-R	TCTGGTGGGCACGATAAGTT	
SVCV ^e^	Detection	SVCV F1	TCTTGGAGCCAAATAGCTCARRTC	[24]
		SVCV R2	AGATGGTATGGACCCCAATACATHACNCAY	

^a^ Megalocytiviruses causing RSIVD, red sea bream iridovirus (RSIV), and infectious nervous necrosis virus (ISKNV), ^b^ nervous necrosis virus, ^c^ viral hemorrhagic septicemia virus, ^d^ hirame rhabdovirus, ^e^ spring viraemia of carp virus.

**Table 2 biology-12-00829-t002:** SYBR Green-based real-time PCR primers used in RSIV and ISKNV gene expression experiments.

Target Gene	Primer Name	Primer Sequence (5′ to 3′)	Condition	Product Size (bp)	Reference
MCP	qM1F	GGCGACTACCTCATTAATGT	95 °C, 10 min (95 °C, 10 s, 60 °C, 1 min) × 40 cycles	141	[29]
	qM1R	CCACCAGGTCGTTAAATGA			
ATPase	qATPase1F	AACAAGCCAGACATGTGTG	95 °C, 10 min, (95 °C, 15 sec, 60 °C, 1 min) × 40 cycles	165	In this study
	qATPase1R	TGTAGTAGCGCACCATGTC			
Polymerase	qPoly1F	GTTTATGGCGGGGGCAAT		136	
	qPoly1R	TGGCCCAGCTGTATGTAGC			
Ring finger protein	qRFP1F	TGTGCAGAGTGCCGTTCAA		161	
	qRFP1R	TGTGTGCCTGTACACACAGT			
Myristoylated membrane protein	qMyristmem1F	TGCGGGACTGGGAGTTG		146	
	qMyristmem1R	TGCCATAAAAGGCGGCC			
β-actin	qDwBactin_F	TAGCCACGCTCTGTCAGGAT		152	
	qDwBactin_R	ACCACCGGTATTGTCATGGA			

## Data Availability

All data presented in this study are included in this article.

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
