# Peer review of "A New Cell Line Derived from the Caudal Fin of the Dwarf Gourami (Trichogaster lalius) and Its Susceptibility to Fish Viruses"

_biology, 2023, doi:10.3390/biology12060829_

Round 1
Reviewer 1 Report
General comments
Viral diseases pose a threat to both farmed and ornamental fish. Thus, the availability of fish cell lines that support the replication of both old and emerging viral pathogens is always welcome. In this paper, the development of an epithelial cell line from dwarf gourami fin (DGF) was escribed. A number of well known fish viruses efficiently replicate in the DFG cell line. Cytopathic effect CPE, virus titers, microscopy examination and viral genome replication was examined. In summary, although not a big advancement of the field (other cell lines are already available for the replication of iridoviruses) this is a solid and thoroughly done work, that may turn out to be helpful for the study, diagnose and control of viral diseases of ornamental fish species.
Minor issues
1.- A common cell line in studies on fish iridoviruses is the mandarin fish fry (MFF). This cell line could have been taken as a comparative reference for DFG when assessing RSIV / ISKNV replication efficiency.
2.- In Results, some section numbers seem duplicated (3.4, 3.5)
3.- In section 3.8 as well as in Figure 11 caption, it should be stated that those data come from a real-time PCR experiment.
4.- Discussion, p477-478: I would not expect VHSV to replicate at 23ºC.
Author Response
Biology
Manuscript ID: biology-2431254
Title: A New Cell Line Derived from the Caudal Fin of the Dwarf Gourami (Trichogaster lalius) and Its Susceptibility to Fish Viruses
# Reviewer 1
Viral diseases pose a threat to both farmed and ornamental fish. Thus, the availability of fish cell lines that support the replication of both old and emerging viral pathogens is always welcome. In this paper, the development of an epithelial cell line from dwarf gourami fin (DGF) was escribed. A number of well known fish viruses efficiently replicate in the DFG cell line. Cytopathic effect CPE, virus titers, microscopy examination and viral genome replication was examined. In summary, although not a big advancement of the field (other cell lines are already available for the replication of iridoviruses) this is a solid and thoroughly done work, that may turn out to be helpful for the study, diagnose and control of viral diseases of ornamental fish species.
Response: Thanks for your comments. In accordance with your comments, we made our best efforts to improve the quality of manuscript. We fully checked and have enriched the discussion section, improved data sources, and reduced confusion in the content. These changes have strengthened the paper overall. In addition, according to editor’s recommendation, we have revised the materials and methods section to minimize similarity with other papers. All the revised parts have been highlighted in red.
Q1. A common cell line in studies on fish iridoviruses is the mandarin fish fry (MFF). This cell line could have been taken as a comparative reference for DFG when assessing RSIV / ISKNV replication efficiency.
A1. Thanks for your suggestion. To address the replication efficiency of RSIV/ISKNV in DGF cells, we have investigated the propagation of ISKNV in the reported MFF-1 cell line, approximately 107.6-8.4 TCID50/mL. We have included this in our manuscript. Furthermore, the report of SKF-9 was also included to enhance the discussion (L503-510).
Q2. In Results, some section numbers seem duplicated (3.4, 3.5)
A2. Thanks for your comment. We apologize for mistakenly writing the section number in our submitted manuscript. We have corrected this error and have checked the entire manuscript.
Q3. In section 3.8 as well as in Figure 11 caption, it should be stated that those data come from a real-time PCR experiment.
A3. Thanks for your insight. We have revised the section 3.8 and figure 11 legend to explicitly state that the data presented were obtained from a real-time PCR(L369-370).
Q4. Discussion, p477-478: I would not expect VHSV to replicate at 23ºC.
A4. Thanks for your comment, and we also agree with your comment. As per your suggestions, VHSV could not replicate exceeding 23°C. We could identify VHSV replication at 20°C on DGF cells. It is still more studies to elucidate regarding VHSV replication in dwarf gourami. And we think that the sentences in our previous manuscript make some confusion. We apologize for the confusion caused by our previous manuscript. By reconstructing the sentence, we improved the accuracy and understanding of our manuscript (L483-486).

Reviewer 2 Report
Q1: Primers for ISKNV are not shown in Table 1.
Q2: Line164-168. This paragraph implies viruses used in the experiment. The sources of viruses are not mentioned.
Q3: The primary cell culture did not mentioned the tissue block adhesion method. Why is there a similar tissue block in Figure 1A?
Q4: Figure 3: It would be better to change the representative pictures of the chromosome to color. Whether the number of representative pictures for chromosome in DGF cells is 2n=44?
Q5: Figure 4: The ordinate needs to be changed.
Q6: The resolution of some figures should be improved.
Author Response
Biology
Manuscript ID: biology-2431254
Title: A New Cell Line Derived from the Caudal Fin of the Dwarf Gourami (Trichogaster lalius) and Its Susceptibility to Fish Viruses
# Reviewer 2
Response: We appreciate the additional reviewer's feedback on our manuscript. In response, we have revised the table and figures to improve clarity and understanding. Furthermore, we have included virus isolation source information, enhancing the robustness of our study. These revisions address the reviewer's comments and strengthen the overall manuscript. In addition, according to editor’s recommendation, we have revised the materials and methods section to minimize similarity with other papers. All the revised parts have been highlighted in red.
Q1. Primers for ISKNV are not shown in Table 1.
A1. Thanks for your comment. With the understanding that both the conventional PCR primer set ([38]) and the real-time PCR primer set and probe ([23]) can detect and quantify both RSIV and ISKNV, it is indeed appropriate to explicitly indicate their applicability in Table 1. To provide a clearer understanding for readers, we have included the phrase “(RSIV and ISKNV)” in Table 1.
Q2. Line164-168. This paragraph implies viruses used in the experiment. The sources of viruses are not mentioned.
A2. Thanks for your comment regarding the lack of information about the sources of the viruses used in the experiment. In response to your suggestion, we have updated our manuscript to include the details about the isolated host (L165-170).
Q3. The primary cell culture did not mentioned the tissue block adhesion method. Why is there a similar tissue block in Figure 1A?
A3. Thanks for your comment. In our study, we did not use a specific tissue block adhesion method for the primary cell culture. The tissue block (that you mentioned) visible in Figure 1A is considered a small fragment of the caudal fin of dwarf gourami that passed through a cell strainer (pore size, 100 μm) used in this study.
Q4. Figure 3: It would be better to change the representative pictures of the chromosome to color. Whether the number of representative pictures for chromosome in DGF cells is 2n=44?
A4. Thanks for your comment. In response to your suggestion, we have made changes to Figure 3. We have replaced the previous representative pictures of the chromosomes with a colored image. Furthermore, we have now provided numbering to each chromosome in the image to facilitate easier identification.
Q5. Figure 4: The ordinate needs to be changed.
A5. Thanks for your comment. We appreciate your feedback on the ordinate (vertical axis) in the figure. In response to your suggestion, we have made the necessary adjustment to the ordinate of the Figures in Figure 4.
Q6. The resolution of some figures should be improved.
A6. Thanks for your comment. In the revised manuscript, we have afforded to improve the resolution of figures.

Reviewer 3 Report
The manuscript i well-written, the experimental desighn is appropriate, cites relevant literature. The text is easy to read and interesting. Materials and methods section is written in an accurate way. I think it should be accepted at present form.